# Weight Categories, Trajectories, Eating Behavior, and Metabolic Consequences during Pregnancy and Postpartum in Women with GDM

**DOI:** 10.3390/nu16040560

**Published:** 2024-02-18

**Authors:** Sybille Schenk, Yann Ravussin, Alain Lacroix, Dan Yedu Quansah, Jardena J. Puder

**Affiliations:** 1Service of Endocrinology, Diabetes and Metabolism, Department of Medicine, Centre Hospitalier Universitaire Vaudois, 1011 Lausanne, Switzerland; sybille.schenk@chuv.ch; 2Obstetric Service, Department Woman-Mother-Child, Centre Hospitalier Universitaire Vaudois, 1011 Lausanne, Switzerland; dan.quansah@chuv.ch; 3Department of Endocrinology, Metabolism and Cardiovascular System, Medicine Section, University of Fribourg, 1700 Fribourg, Switzerland; yann.ravussin@unifr.ch; 4Institute of Higher Education and Research in Healthcare, University of Lausanne, 1010 Lausanne, Switzerland; alain.lacroix@bluewin.ch

**Keywords:** diabetes, pregnancy, obesity, gestational weight gain, intuitive eating

## Abstract

Pre-pregnancy overweight and obesity are associated with increased risk for adverse outcomes, such as gestational diabetes mellitus (GDM). This study investigated weight trajectories, eating behaviors, and metabolic consequences in women with GDM during pregnancy and postpartum according to pre-pregnancy BMI. We prospectively included 464 women with GDM. Intuitive eating (Intuitive Eating Scale-2 questionnaire), gestational weight gain (GWG), postpartum weight retention (PPWR) at 6–8 weeks and 1-year postpartum, and glucose intolerance (prediabetes and diabetes) at 1-year were assessed. Women with obesity (WOB) had lower GWG but gained more weight in the postpartum (*p* < 0.0001). PPWR at 1-year did not differ across BMI categories (*p* = 0.63), whereas postpartum weight loss was most pronounced in women with normal weight (*p* < 0.0001), and within this category, in their lowest tertile (*p* < 0.05). Intuitive eating was not linked to perinatal weight changes but differed among BMI categories. PPWR predicted a 2.5-fold increased risk of glucose intolerance at 1-year independent of pre-pregnancy BMI (*p* < 0.001), and the adverse metabolic impact of PPWR was most pronounced in WOB with odds of increased risk of glucose intolerance 8.9 times higher (95% CI 2.956;26.968). These findings suggest an adaptive capacity to relatively rapid weight changes in the perinatal period that is less present with higher BMI.

## 1. Introduction

The increasing prevalence of obesity is a public health concern [1]. Obesity is associated with dysregulated eating behavior [2,3] and is related to metabolic complications such as increased risk of type 2 diabetes, cardiovascular diseases, and increased mortality [4]. In the perinatal period, pre-pregnancy overweight (25.0–29.9 kg/m^2^) and obesity (≥30 kg/m^2^) are associated with adverse outcomes for the mother and the offspring [5], including, for the mother, an increased risk of gestational diabetes mellitus (GDM) [6]. GDM, defined as any degree of glucose intolerance first diagnosed during pregnancy that does not fulfill the criteria for diabetes [7], is influenced by the degree of gestational weight gain (GWG) [8]. Approximately 50% of all pregnant women have excessive gestational weight gain (EGWG), using the GWG recommendations of the National Academy of Medicine (NAM, previously IOM) [9,10]. Both increased GWG and EGWG are associated with higher postpartum weight retention (PPWR) [11,12], which is the most relevant risk factor for metabolic complications, particularly diabetes for these women [13]. In metabolically high-risk women with GDM, EGWG may further exacerbate maternal insulin resistance and increase the risk of perinatal complications [14]. Similarly, EGWG is associated with increased PPWR [15] and a more adverse metabolic profile in the postpartum in this population [16]. Indeed, PPWR is a predictor of future diabetes [17], and in women after GDM, even a modest reduction of 1 kg in PPWR is associated with a 16% lower risk of diabetes [18]. 

Weight gain or obesity can be influenced by eating behavior, which relates to eating practices, food choices and motives, feeding practices, dieting, and eating-related problems [3,19,20,21]. Adaptive eating behavior, such as intuitive eating (IE), characterized by eating in response to physiological hunger and satiety cues rather than external and/or emotional cues [22,23], is associated with lower weight outside of pregnancy [24,25]. In metabolically high-risk women with GDM, IE has been associated with lower pre-pregnancy BMI, lower weight during pregnancy, and lower weight during the first postpartum year [26,27]. In the general population, feelings of fullness and satiety have been associated with lower PPWR [28]. Overall, the relationship between IE and BMI as well as weight changes needs further investigation given the limited existing data, particularly in the perinatal period and in more high-risk populations [29].

Weight trajectories also differ across pre-pregnancy BMI categories [30]. EGWG is not exclusive to a specific BMI category [31] but is more frequent in women with overweight (WOW) and obesity (WOB) [9]. Women with underweight (WUW) and normal weight (WNW) generally experience less PPWR, while WOW and WOB may encounter challenges in postpartum weight management and face an elevated risk of long-term obesity [32,33]. In addition, the metabolic risk related to PPWR might be most pronounced in women with higher pre-pregnancy BMI [34,35]. However, few studies have comprehensively studied the differences in weight trajectories during and after pregnancy related to their pre-pregnancy BMI [30] and these studies are lacking in women with GDM.

Throughout pregnancy, women undergo a range of physical and metabolic changes that aim to safeguard fetal growth and development. These changes are closely linked to both their pre-pregnancy weight, nutritional status, and the amount of weight they gain during pregnancy. Pregnancy is a unique natural metabolic phase that necessitates initial weight gain, followed by weight loss after delivery [12].

Data on the effects of rapid weight changes in humans are scarce [36,37]. In healthy rodents, rapid weight gain followed by weight loss does not seem to have long-term effects on metabolic health, unlike a slower weight gain for which the compensatory responses during and after weight loss might differ [38]. The regulation of body weight involves central compensatory responses triggered amongst others by changes in fat mass [36]. Moreover, adipocytes can retain a transcriptional memory of obesity that persists even after weight loss [39]. It remains to be elucidated how these changes affect metabolic health, especially in the context of relatively rapid changes during pregnancy and the postpartum, and to what extent this depends on the initial pre-pregnancy weight and thus the state of the adipocytes.

To fill the gap in the literature, we investigated weight trajectories, eating behaviors, and metabolic consequences during pregnancy up to 1-year postpartum according to pre-pregnancy BMI in women with GDM. We investigated (a) the differences in weight trajectories according to pre-pregnancy BMI categories, (b) the relationship between eating behavior and BMI categories and weight trajectories, and (c) the impact of pre-pregnancy BMI categories and weight trajectories on future metabolic health.

## 2. Materials and Methods

### 2.1. Participant Consent and Recruitment

This prospective cohort followed women with GDM during pregnancy up to 1-year postpartum between 2013 and 2022. Pregnant women with GDM were invited to participate in the study at the diabetes in pregnancy clinic of the Lausanne University Hospital, where patients are referred from both the antenatal care clinic and obstetricians in private practice. The Human Research Ethics Committee of the Canton de Vaud approved the study protocol (326/15).

### 2.2. Exclusion and Inclusion Criteria

Out of a cohort of 1430 women assessed for eligibility, 287 were excluded (Figure 1). The main reason for exclusion was gestational age (GA). Participants whose first GDM visit was too early (<20 weeks GA) or too late (>32 weeks GA) were excluded. Participants with type 1 or type 2 diabetes, bariatric surgery, OGTT results who did not meet the GDM criteria, or a participation in an intervention study were also excluded. Regarding the inclusion criteria, of the 1143 eligible women, those with valid data for weight at the 3 time points (pre-pregnancy, at 6–8 weeks, and at 1-year postpartum) were included. Overall, 464 women with valid data at all three time points were included in the final analysis.

### 2.3. GDM Diagnosis and Patient Follow-Up 

Women were generally diagnosed with GDM at 24–28 weeks GA if one of the following criteria were met during a 75 g oral glucose tolerance test (OGTT) (fasting plasma glucose (FPG) ≥ 5.1 mmol/L, 1 h glucose ≥ 10.0 mmol/L, or 2 h glucose ≥ 8.5 mmol/L) in accordance with the International Association of Diabetes and Pregnancy Study Groups (IADPSG) and the American Diabetes Association (ADA) guidelines [7,40]. All women were followed according to the current ADA guidelines [7]. The women had regular appointments every 1–3 weeks with a physician or a diabetes-specialist nurse and with a dietician after the GDM diagnosis. The women received advice on adherence to GWG recommendations based on the NAM guidelines: recommended weight gain was 12.5–18 kg if pre-pregnancy BMI was <18.5 kg/m^2^, 11.5–16 kg if it was 18.5–24.9 kg/m^2^, 7.0–11.5 kg if it was 25.0–29.9 kg/m^2^, and 5–9 kg if pregnancy BMI was >30 kg/m^2^ [10]. Insulin treatment was initiated when glucose values were above targets more frequently than once per week for given period of the day (fasting glucose ≥5.3 mmol/l or 2 h postprandial glucose ≥7 mmol/L), in accordance with Swiss guidelines [41,42]. The insulins most frequently used were NPH and aspartate. The postpartum follow-up visits at both 6–8 weeks postpartum and 1-year postpartum included anthropometric measures, an assessment of the glycemic control, and counseling on lifestyle changes based on the laboratory and anthropometric results without introducing any medications before 1-year postpartum. 

### 2.4. Measures

#### 2.4.1. Sociodemographic and Medical Characteristics

Data on maternal socio-demographic characteristics including age, ethnicity and educational level, social support, previous history of GDM, family history of diabetes, parity and gravida, GA, tobacco use, or alcohol consumption during pregnancy were collected during the first GDM visit or during clinical follow-up visits. Data on breastfeeding status were assessed during both postpartum visits. 

#### 2.4.2. Intuitive Eating Behavior

We assessed IE with a 14-item self-report questionnaire consisting of “eating for physical rather than emotional reasons” (EPR, 8 items) and the “reliance on hunger and satiety cues” (RHSC, 6 items) subscales of the French Intuitive Eating Scale-2 (IES-2) [43] with scores ranging between 1 and 5 for each subscale. A higher score of the EPR subscale reflects eating as a response to hunger, and a lower score as a response to emotional distress, whereas a higher score of the RHSC subscale signifies increased reliance on internal cues, and a lower score a reduced ability to regulate food intake based on internal cues. The use of IES-2 in our cohort has been previously described elsewhere [26,27,44].

#### 2.4.3. Anthropometric Measures

Pre-pregnancy weight was extracted from the participants’ medical charts or, very rarely, was self-reported if missing. During the first GDM visit, body weight and height were measured to the nearest 0.1 kg and 0.1 cm, respectively, with an electronic scale (Seca^®^). We also measured participants’ weight at the end of pregnancy, at 6–8 weeks, and at the 1-year postpartum visit. We calculated BMI as the ratio of weight in kilograms to the square of height in meters (kg/m^2^). We categorized women into normal (<24.9 kg/m^2^), overweight (25–29.9 kg/m^2^), or obese (≥30 kg/m^2^), according to their pre-pregnancy BMI. We calculated GWG as the difference between the weight at the end of pregnancy and the weight before pregnancy. EGWG was calculated based on the NAM recommendations. We defined PPWR as the difference between the weight at 6–8 weeks or 1-year postpartum (≥0.1 kg) and the weight before pregnancy. We also calculated the difference between the weight at the end of pregnancy and at 6–8 weeks postpartum.

#### 2.4.4. Assessment of Glycemic Control Variables

At 6–8 weeks postpartum, we measured HbA1c and performed a 75 g OGTT to measure FPG and 2 hr glucose. At 1-year postpartum, only FPG and HbA1c were measured. In the postpartum, we defined abnormal glucose tolerance (FPG ≥ 5.6 mmol/L or HbA1c ≥ 5.7% or 2 h glucose ≥ 7.8 mmol/L), prediabetes (FPG 5.6–6.9 mmol/L or HbA1c 5.7–6.4% or 2 h glucose 7.8–11.0 mmol/L), or diabetes (FPG ≥ 7.0 mmol/L, 2 h glucose ≥ 11.1 mmol/L or HbA1c ≥ 6.5%), according to the ADA criteria [7]. Prediabetes and diabetes were pooled together and referred to as “glucose intolerance”. Women were not allowed to breastfeed during the OGTT, and those who reported for testing without an overnight fast had their OGTT rescheduled.

### 2.5. Statistical Analysis 

All analyses were performed with Stata/SE 15.1 (StataCorp LLC, College Station, TX, USA). All variables were normally distributed. Descriptive variables were presented as either means (±standard deviation) or in percentages (%) where appropriate. ANOVA (continuous variables) or chi-squared tests (categorical variables) were used to determine the differences in weight trajectories, eating behaviors, and the prevalence of glucose intolerance at 1-year postpartum (Appendix A) according to the three pre-pregnancy BMI categories (WNW, WOW, WOB), as well as to compare weight trajectories according to tertiles within WNW (based on pre-pregnancy BMI, Appendix A).

*t*-tests were used to determine the weight change between 6–8 weeks and 1-year postpartum according to presence or absence of PPWR and to compare PPWR at 1-year postpartum according to the presence or absence of EGWG (Appendix A). We performed logistic regression analyses to determine the impact of GWG, postpartum weight changes, and PPWR on abnormal glucose tolerance at 1-year postpartum. All statistical significance was two-sided and accepted at *p* < 0.05.

## 3. Results

Table 1 shows the summary of the socio-demographic and medical characteristics of study participants. Their mean age and pre-pregnancy BMI were 33.2 ± 5.1 years and 26.3 ± 5.7 kg/m^2^, respectively. Of the 464 women, 49.6% had a normal weight (n = 230, including 6% WUW), 29.7% had overweight (n = 138), and 20.7% had obesity (n = 96) before pregnancy. There were no differences in age, pre-pregnancy BMI, or weight at the first GDM visit between the complete (n = 464) and non-complete (n = 679) datasets. Of note, the rates of breastfeeding at both 6-8 weeks and 1-year postpartum did not differ between the BMI categories (*p* = 0.238 and *p* = 0.173, respectively). 

Table 2 and Figure 2 describe the weight trajectories throughout pregnancy and up to 1-year postpartum, according to pre-pregnancy BMI categories. The WOB GWG average (8.3 ± 7.6 kg) was significantly lower than the WNW (13.6 ± 5.6 kg, *p* < 0.0001) and the GWG of the WOW (13.2 ± 5.6 kg, *p* < 0.0001; p for overall differences *p* < 0.0001). Mean GWG was similar in the WNW and WOW. Additionally, the GWG was not significantly different between the women who did not receive insulin (11.9 ± 5.9 kg) and those who did receive insulin (13.0 ± 6.9 kg) (*p* > 0.05) during pregnancy. On the other hand, the proportion of EGWG was highest in the WOW (51.5%) compared to both WNW (23.9%) and WOB (35.4%) (*p* < 0.0001). PPWR at 1-year postpartum did not differ according to BMI category. While there were no differences in weight changes between the end of pregnancy and 6–8 weeks postpartum according to pre-pregnancy BMI groups (*p* = 0.354), there were significant differences between the 6–8 weeks and 1-year postpartum: Specifically, WNW lost a mean of 2.3 ± 4.9 kg whereas WOB gained a mean of 3.5 ± 7.8 kg (overall *p* < 0.0001).

In addition, we investigated the relationship between GWG and PPWR. GWG was positively associated with PPWR at 1-year (β: 0.46 kg, 95% CI 0.369;0.546, *p* < 0.0001), i.e., for every kg gained in pregnancy, the PPWR increased by a mean of 0.46 kg. We compared PPWR at 1-year according to EGWG categories (Appendix A). Overall, women with EGWG had a higher PPWR at 1-year compared to women without EGWG (6.0 ± 7.4 kg vs. 2.2 ± 5.5 kg; *p* < 0.001). When we adjusted for GWG, WOB had a higher mean PPWR of 2.6 kg (95%CI 1.09–4.15, *p* < 0.0001) at 1-year postpartum than WNW and of 2.3 kg (95%CI 0.29–4.36, *p* < 0.0001) than WOW. These results did not change when we further adjusted for age.

Table 3 describes the differences in weight changes between 6–8 weeks and 1-year postpartum according to PPWR. Overall, women without PPWR at 1-year postpartum lost 4.5 ± 5.8 kg. In all pre-pregnancy BMI groups, women without PPWR at 1-year postpartum lost weight between the early and late postpartum, while in those with PPWR, only the WNW lost weight (−1.43 ± 4.68 kg, *p* < 0.0001).

To further investigate metabolically healthy women, we divided WNW into tertiles based on their pre-pregnancy BMI (Appendix A). In these WNW, GWG did not differ amongst these tertiles (overall *p* = 0.737). However, between 6–8 weeks to 1-year postpartum, women in the lowest tertile lost twice as much weight than those in the other tertiles (overall p for differences between tertiles = 0.018). Similarly, PPWR at 1-year was least in the lowest tertile (2.3 ± 4.1 kg), compared to the middle and highest NW tertiles (4.5 ± 5.1 kg, 3.1 ± 5.6 kg, overall *p* = 0.025 for differences between tertiles, see Appendix A for more details).

We also investigated the differences in eating behavior according to pre-pregnancy BMI groups (Table 4). We found that during pregnancy, mean scores of the EPR subscale of the IES-2 differed between BMI categories (higher when BMI was lower, *p* = 0.024), while this was not observed for the RHSC subscale (*p* = 0.207). At 1-year postpartum, differences were observed for both the EPR and the RHSC subscale (*p* ≤ 0.028). While the IES-2 scores differed across pre-pregnancy BMI categories, they were not associated with changes in GWG, postpartum weight changes or PPWR, either in the total population or in the BMI-subgroups (See Table 4 for more details).

Table 5 describes the impact of GWG, weight changes, and PPWR on abnormal glucose intolerance at 1-year postpartum according to pre-pregnancy BMI categories. In this cohort, 36% of women had prediabetes whereas 2.8% had diabetes at 1-year postpartum (See Appendix A for details according to BMI category). While GWG was not related to abnormal glucose tolerance at 1-year postpartum, the presence of PPWR at 1-year postpartum was associated with a 2.47-fold increased risk of abnormal glucose tolerance, independent of age and pre-pregnancy BMI (95% CI 1.5–3.9, *p* < 0.001). This association was especially pronounced in WOB (OR 8.9, 95% CI 2.9–26.9, *p* < 0.0001). Similarly, women who gained weight between the early and the late (1-year) postpartum, had a 1.74-fold (95% CI 1.19–2.54, *p* = 0.004) increased risk of abnormal glucose tolerance. The latter association was attenuated after adjusting for age and pre-pregnancy BMI.

## 4. Discussion

In this cohort of women with GDM, weight trajectories during pregnancy up to 1-year postpartum differed according to pre-pregnancy BMI categories. Although WOB had the lowest GWG, their weight increased between 6–8 weeks and 1-year postpartum, leading to a similar mean PPWR across pre-pregnancy BMI categories. When adjusted for GWG, PPWR was highest in WOB (since GWG was lowest in that group). Eating behavior differed according to BMI categories but had no impact on perinatal weight trajectories. The presence of PPWR at 1-year was associated with a 2.47-fold increased risk of abnormal glucose tolerance in all women and an 8.9-fold increased risk in WOB. In addition, weight gain in the postpartum was associated with increased risk of glucose intolerance. These results hint to a higher plasticity regarding overall weight changes in the perinatal period in women with lower BMI and demonstrate the particularly pronounced role of PPWR in WOB.

So far, no previous study has investigated weight trajectories and potential contributing factors such as eating behavior throughout pregnancy and up to 1-year postpartum in women with GDM and stratified by pre-pregnancy BMI categories. We observed that WOB had lower GWG compared to WNW and WOW, the latter having similar GWG. Our findings align with previous studies in women with or without GDM [45,46,47], which found that WOB had the lowest GWG compared to other BMI categories. Despite this, women in all BMI categories had an EGWG at the first GDM visit at around 28 weeks of gestational age (mean EGWG ranged between 0.74 kg for all WOB and 3.25 kg for all WOW). However, only WOW remained with EGWG throughout the entire pregnancy, after GDM diagnosis.

PPWR at 1-year did not differ across BMI categories, which is consistent with most [48,49], but not all studies [11]. Differences between BMI categories may be attributed to various factors influencing PPWR, such as discrepancies in EGWG, presence or not of breastfeeding, parity, differences in dietary intakes, the use of insulin during pregnancy, and a lower education status [15]. In our cohort, GWG was positively associated with PPWR at 1-year, in line with the literature [50,51]. Specifically, each kilogram gained in pregnancy increased PPWR by 0.45 kg. As previously shown [48,49], women with EGWG had a higher PPWR at 1-year than those without EGWG, and this was most pronounced in WOB. Data regarding differences in PPWR according to BMI categories are scarce in women with GDM. However, in a meta-analysis examining pre-pregnancy BMI’s impact on PPWR in the general pregnant population found that as the BMI increased, the average PPWR decreased [11]. Overall, these findings indicate the importance of pre-pregnancy BMI and of strategies for pre-pregnancy obesity prevention or treatment in addition to reducing GWG. 

Pre-pregnancy BMI categories impacted postpartum weight trajectories: WNW lost a mean of 2.3 kg weight, WOW lost 0.6 kg while WOB gained a mean of 3.5 kg. In a study analyzing weight trajectories in women without GDM, WOW or WOB had higher odds of initial weight loss followed by slight regain [52]. Other studies in women without GDM also showed higher odds of weight gain in the postpartum for WOW or WOB [53,54]. The observation of different postpartum weight changes raises the question of a different plasticity to perinatal weight fluctuations and a higher capacity to lose weight in the postpartum in women with lower BMI before pregnancy.

To study this more in detail, we categorized the WNW into tertiles. Despite similar GWG, postpartum weight loss was most pronounced in women in the lowest tertile. These women lost more weight between 6–8 weeks and 1-year postpartum than those in higher tertiles, and their PPWR was lowest. This underlies the possibility that women with lower pre-pregnancy BMI may have an advantage in responding to weight changes during and after pregnancy and that this plasticity may explain why WNW might be more adept at losing the excess weight gain (i.e., following short periods of overfeeding such as during holidays or certain injuries that require immobility, these women may be more adept at losing the excess weight gain). Research on weight recovery post-overfeeding in rodent models showed interesting insights into weight regulation [38]. Overfeeding-induced obesity during a short period in healthy mice induced a rapid weight gain that could represent an analogy to weight changes in pregnancy. In these mice, this was followed by an anorectic response that led to a return to their initial body weight. There might be a similar process occurring in humans regarding the relative short-term weight changes in the perinatal period, particularly in healthy women with lower pre-pregnancy BMI. This physiological capacity to lose the excess weight may be hindered in women with higher pre-pregnancy BMIs. In humans with pre-existing overweight or obesity, the ability to return to a previously lower body weight and maintain this loss might be complicated on different levels. After weight loss, a sustained increase in activity does not consistently elevate the total energy expenditure due to potential compensatory mechanisms [55]. The degree of energy compensation varies based on body composition, and these compensatory mechanisms are more pronounced in individuals with higher initial body fat, making longer-term sustained fat loss more challenging in this population [55].

Differences found in responses to postpartum weight loss and PPWR between women with distinct pre-pregnancy BMIs might be occurring due to peripheral adaptations such as adipocyte functioning and/or in central regulation [36]. Adipocytes of previously obese mice after weight loss can exhibit a memory including different predispositions to store fat [39]. Regarding the central regulation of body weight, there are potential differences between previously obese and non-obese humans in the regulation of hunger and satiety, the control of appetite, or the brain’s responses to changes in body weight. In addition, hormonal factors such as increased brain insulin resistance with implications on the homeostatic set point, modulating peripheral energy metabolism and weight loss might also play a role, especially in women with GDM and even more in the WOB [56,57]. 

We investigated both potential causes of different weight trajectories and their metabolic implications in women with GDM within the same cohort. Although IE scores differed among BMI categories in our cohort, IE behavior was not associated with perinatal weight trajectories including GWG, weight changes between 6-8 weeks and 1-year postpartum, or PPWR at 1-year. According to Paterson et al. [58], IE may influence GWG, especially the unconditional permission to eat (UPE) subscale. In our study, however, we did not investigate the UPE subscale as all women had seen a dietician after their GDM diagnosis, and this was thought to have an impact on UPE.

Although GWG was not associated with an increased risk of abnormal glucose tolerance at 1-year postpartum, the presence of PPWR at 1-year was associated with a 2.47-fold increased risk, independent of age and pre-pregnancy BMI. Weight gain between 6-8 weeks and 1-year postpartum also increased the risk for abnormal glucose tolerance, but only in the unadjusted model. The observed adverse metabolic profile as early as 1-year postpartum in women who do not lose all the pregnancy weight or even gain weight during this period highlights the importance of postpartum weight management. Similarly, adverse cardiometabolic profiles can emerge as early as 1-year postpartum in women who do not lose weight between 3 and 12 months after delivery [59]. In accordance with another study [60], the impact of PPWR was more pronounced in WOB, as the presence of PPWR increased their risk of glucose intolerance by 8.9-fold. 

Our findings underscore the importance of early intervention to improve adverse outcomes in the postpartum. The observed postpartum weight gain and the impact of PPWR on adverse metabolic health, especially in the WOB, necessitates an increased level of clinical vigilance and a multidisciplinary approach involving endocrinologists, dietitians, and other healthcare professionals. The identification of distinct BMI trajectories shows the need for personalized care strategies. Our findings could help develop targeted metabolic health strategies, based on pre-pregnancy BMI to improve adverse outcomes for high-risk women with a particular focus on the postpartum, as this seems to be a critical window [15,61].

This study has several strengths. This longitudinal cohort investigated an understudied topic in women with GDM. To our knowledge, this is the first study to investigate weight trajectories during pregnancy and up to 1-year postpartum in women with GDM focusing on different BMI categories and including eating behavior and metabolic consequences. However, the associations found may be correlational rather than causal, despite our prospective design. We obtained weight before pregnancy from patients’ medical chart when available; otherwise, we relied on self-reported pre-pregnancy weight. However, there was a strong correlation between clinically measured weight during and after pregnancy with pre-pregnancy weight (r = 0.94). As there were only few WUW (n = 14) (<18.5 kg/m^2^), we included them in the NW category. In our cohort, only two subscales of the French IES-2 were used; we did not investigate the UPE subscale as well as an overall IES-2 total score, limiting the interpretation of our results regarding IE. Moreover, another limitation in this study is the lack of assessment of data regarding energy and macronutrients as this could influence GWG. The impact of physical activity on GWG was also not assessed in this cohort. 

## 5. Conclusions

In this cohort of GDM women, a lower GWG predicted a reduced PPWR. However, pre-pregnancy BMI had a significant impact on the perinatal weight trajectories. WOB had the lowest GWG, but, in contrast to WNW and WOW, they gained weight in the postpartum period. These postpartum weight changes were not associated with intuitive eating behaviors. Postpartum weight loss was most pronounced in WNW. This could suggest an adaptive capacity to the body’s response to relatively rapid weight changes in the perinatal period, which is especially present or more pronounced when pre-pregnancy BMI is lower. PPWR or postpartum weight gain increased the risk of glucose intolerance, and the adverse metabolic impact of PPWR was most deleterious in WOB. These findings emphasize the importance of monitoring and managing weight gain before, during, and after pregnancy to reduce the risk of adverse health outcomes, especially in WOB.

## Figures and Tables

**Figure 1 nutrients-16-00560-f001:**
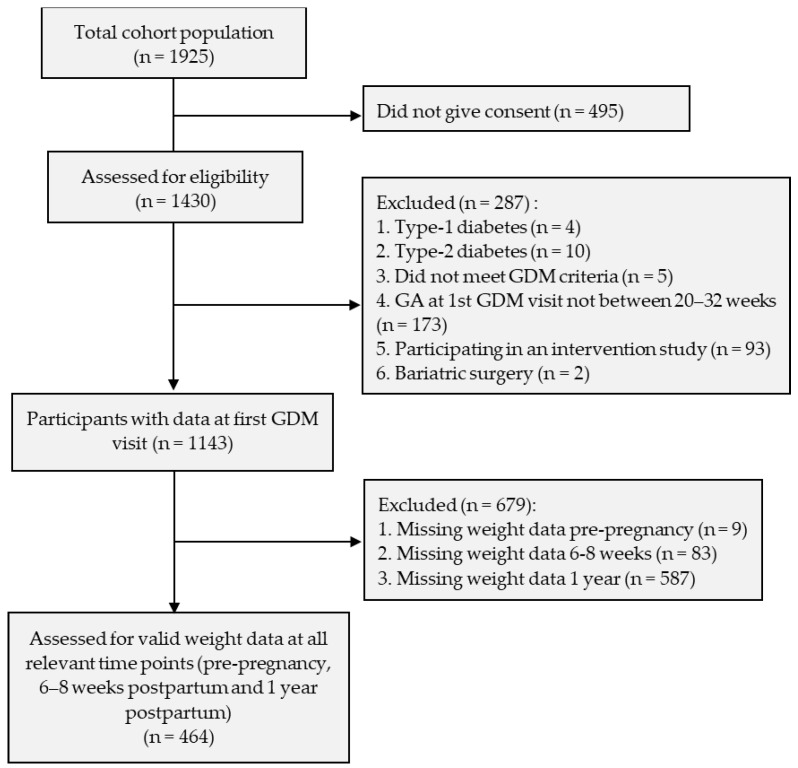
Flow-chart of participant selection in the study.

**Figure 2 nutrients-16-00560-f002:**
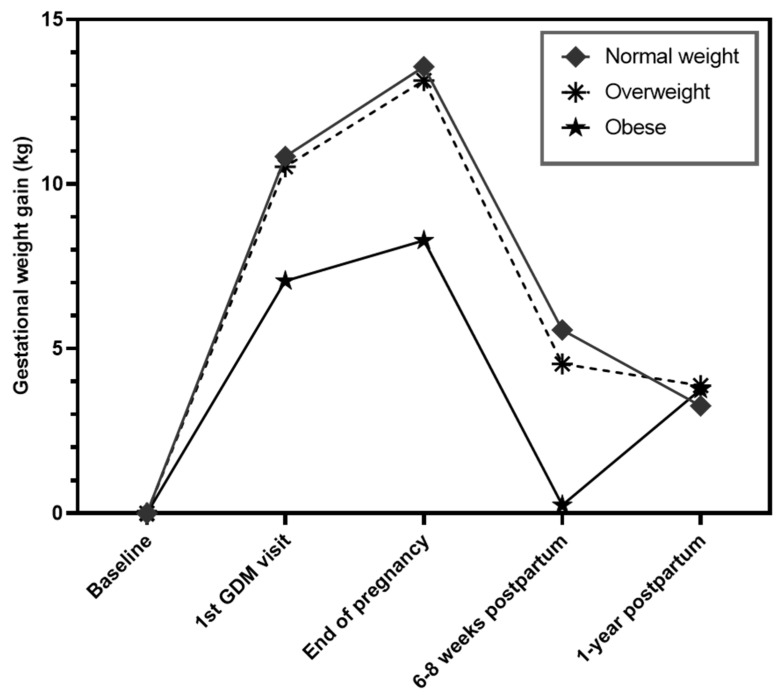
Weight trajectories during and after pregnancy in women with GDM according to normal (◆), overweight (🞿), and obese (★) pre-pregnancy BMI. Baseline denotes the changes starting at the beginning of pregnancy.

**Table 1 nutrients-16-00560-t001:** Socio-demographic characteristics of the participants (n = 464).

Variables	Mean ± SD	n (%)
**Age (years)** **Gestational age at 1st GDM visit (weeks)**	33.21 ± 5.12 28.71 ± 1.90	
**Gestational age at delivery (weeks)**	38.75 ± 1.90	
**Pre-pregnancy BMI (kg/m^2^) ^a^**	26.26 ± 5.69	
**Educational level (n = 345) ^b^**		
Lack of schooling		5 (1.45)
Compulsory school unachieved		28 (8.12)
Compulsory school achieved		66 (19.13)
High school		33 (9.57)
General and vocational education		77 (22.32)
University		136 (39.42)
**Ethnicity**		
Switzerland		130 (28.02)
Europe + North America		164 (35.34)
Africa		84 (18.10)
Asia + Oceania		66 (14.22)
Latin America		12 (2.59)
Others		8 (1.72)
**Social support ^c^**		
No		46 (9.91)
Yes		418 (90.09)
**Smoking during pregnancy (n = 446) ^d^**		
No		386 (86.55)
Yes		60 (13.45)
**Alcohol intake during pregnancy (n = 394) ^e^**		
No		367 (93.15)
Yes		27 (6.85)
**Previous history of GDM**		
No		446 (96.12)
Yes		18 (3.88)
**Family history of diabetes (n = 441) ^f^**		
First degree		167 (37.87)
Second degree		100 (22.68)
No		174 (39.45)
**Gravida**		
1		151 (32.54)
2		140 (30.17)
≥3		173 (37.28)
**Parity**		
0		229 (49.35)
1		141 (30.39)
≥2		94 (20.26)
**Breastfeeding at hospital discharge (n = 439) ^g^**		
No		22 (5.01)
Yes		417 (94.99)
**Breastfeeding at 6-8 weeks postpartum (n = 437) ^h^**		
No		83 (18.99)
Yes		354 (81.01)
**Breastfeeding at 1-year postpartum (n = 401) ^i^**		
No		276 (68.83)
Yes		125 (31.17)

Data are shown as mean ± standard deviation or numbers and percentages. ^a^ Women with a pre-pregnancy BMI <18.5 kg/m^2^ were included in the Normal pre-pregnancy BMI group due to a small number (n = 14). ^b^ 119 participants had missing data on educational level. ^c^ Social support denotes living with partner or with support. ^d^ 18 participants had missing data on smoking during pregnancy. ^e^ 70 participants had missing data on alcohol intake. ^f^ 23 participants had missing data on family history of diabetes. ^g^ 25 participants had missing data on breastfeeding at hospital discharge. ^h^ 27 participants had missing data on breastfeeding at 6–8 weeks postpartum. ^i^ 63 participants had missing data on breastfeeding at 1-year postpartum.

**Table 2 nutrients-16-00560-t002:** Weight trajectories according to pre-pregnancy BMI categories.

Variables	Total (n = 464)	(A) WNW (n = 230)	(B) WOW (n = 138)	(C) WOB (n = 96)	Overall *p*-Value	*p*-Value B vs. A	*p*-Value C vs. A	*p*-Value B vs. C
**Pre-pregnancy BMI**	26.26 ± 5.69	22.02 ± 1.99	27.07 ± 1.31	35.27 ± 4.40	<0.0001	<0.0001	<0.0001	<0.0001
**GWG**	12.39 ± 6.39	13.58 ± 5.64	13.15 ± 5.63	8.29 ± 7.58	<0.0001	1.000	<0.0001	<0.0001
**EGWG (yes: n, %)**	160 (34.48)	55 (23.91)	71 (51.45)	34 (35.42)	<0.0001	N/A	N/A	N/A
**∆ Weight ** **between end of pregnancy and 6–8 weeks postpartum**	8.17 ± 3.13	7.99 ± 2.84	8.50 ± 3.21	8.10 ± 3.69	0.354	0.463	1.000	1.000
**PPWR at 6–8 weeks postpartum**	4.16 ± 6.04	5.57 ± 5.37	4.54 ± 5.22	0.25 ± 6.96	<0.0001	0.288	<0.0001	<0.0001
**PPWR at 1-year ** **postpartum**	3.54 ± 6.46	3.26 ± 5.03	3.88 ± 7.06	3.76 ± 8.38	0.629	1.000	1.000	1.000

Data are shown as means ± standard deviation unless noted otherwise. GWG denotes gestational weight gain, EGWG denotes excess gestational weight gain, PPWR denotes postpartum weight retention. *p*-values are derived from ANOVA for continuous or from Chi-square test for categorical variables. N/A denotes not applicable.

**Table 3 nutrients-16-00560-t003:** Weight changes between 6–8 weeks and 1-year postpartum according to PPWR.

Participants	Weight Changes between 6–8 Weeks and 1-Year Postpartum			*p*-Values for Differences between Groups
	Presence of PPWR at 1-Year Postpartum	Absence of PPWR at 1-Year Postpartum	Differences between Groups *	
**Total, kg (n = 464)**	0.76 ± 5.78	−4.49 ± 5.79	−5.25 ± 0.61	<0.0001
**WNW, kg (n = 230)**	−1.43 ± 4.68	−4.85 ± 4.69	−3.41 ± 0.71	<0.0001
**WOW, kg (n = 138)**	0.93 ± 4.50	−5.37 ± 5.95	−6.30 ± 0.96	<0.0001
**WOB, kg (n = 96)**	6.04 ± 6.55	−2.64 ± 7.33	−8.67 ± 1.52	<0.0001

Data are shown as means ± standard deviation unless noted otherwise. WNW denotes women with normal pre-pregnancy BMI, WOW denotes women with overweight pre-pregnancy BMI, WOB denotes women with obese pre-pregnancy BMI and PPWR denotes postpartum weight retention. *p*-values are derived from *t*-tests. * Unadjusted differences. For all women together, when adjusting for BMI categories, the difference in weight changes in the postpartum between those with and without PPWR was 5.41 kg (95% CI 4.31;6.52, *p* < 0.0001).

**Table 4 nutrients-16-00560-t004:** Eating behavior according to pre-pregnancy BMI categories.

Variables	Total (n = 245)	(A) WNW (n = 134)	(B) WOW (n = 64)	(C) WOB (n = 47)	Overall *p*-Values	*p*-Value B vs. A	*p*-Value C vs. A	*p*-Value B vs. C
**IE in pregnancy** EPR score	3.85 ± 0.86	3.98 ± 0.84	3.77 ± 0.88	3.60 ± 0.84	0.024	0.332	0.028	0.897
**IE in pregnancy** RHSC score	3.51 ± 0.85	3.60 ± 0.83	3.41 ± 0.87	3.40 ± 0.87	0.207	0.414	0.517	1.000
**IE at 1-year pp** EPR score	3.73 ± 0.95	3.92 ± 0.85	3.62 ± 1.04	3.29 ± 0.95	0.004	0.144	0.005	0.355
**IE at 1-year pp** RHSC score	3.47 ± 0.97	3.64 ± 0.96	3.27 ± 0.95	3.28 ± 0.91	0.028	0.047	0.220	1.000

Data are shown as means ± standard deviation unless noted otherwise. IE denotes intuitive eating, EPR denotes eating for physical rather than emotional reasons, RHSC denotes reliance on hunger and satiety cues, WNW denotes women with normal pre-pregnancy BMI, WOW denotes women with overweight pre-pregnancy BMI, WOB denotes women with obese pre-pregnancy BMI. *p*-values are derived from ANOVA for continuous.

**Table 5 nutrients-16-00560-t005:** Impact of GWG, postpartum weight changes, and PPWR on glucose intolerance at 1-year postpartum.

	Model 1			Model 2		
Variables	OR	95% CI	*p*-Value	OR	95% CI	*p*-Value
**GWG**	0.98	0.95–1.01	0.192	0.99	0.97–1.03	0.952
**PPWR at 1-year ** **postpartum**	2.32	1.46–3.68	<0.001	2.47	1.54–3.95	<0.001
**Weight gain between 6-8 weeks and 1-year postpartum**	1.74	1.19–2.54	0.004	1.35	0.90–2.03	0.152
**PPWR at 1-year and weight gain between 6–8 weeks to 1-year postpartum VS all others**	2.07	1.41–3.06	<0.001	1.71	1.14–2.57	0.009

Data are shown as odds ratios and 95% confidence intervals unless noted otherwise. Model 1: unadjusted. Model 2: adjusted for age and pre-pregnancy BMI (continuous variable). GWG denotes gestational weight gain and PPWR denotes postpartum weight retention. *p*-values are derived from logistic regressions.

## Data Availability

The data presented in this study are available on request from the corresponding author. The data are not publicly available because they are clinical data maintained and kept in a secure server at the Lausanne University Hospital.

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
