# Peer review of "Weight Categories, Trajectories, Eating Behavior, and Metabolic Consequences during Pregnancy and Postpartum in Women with GDM"

_nutrients, 2024, doi:10.3390/nu16040560_

Round 1

Reviewer 1 Report

Comments and Suggestions for Authors

I read with great interest this paper. This prospective study is well-designed with large numbers of patients. This is a nowel study that adds new data to current common knowledge regarding an adaptive capacity to relatively rapid weight changes in the perinatal period.

I have some comments to improve the paper.

1. In the text authors indicated that the main reasons for exclusion included too early (<24 weeks gestational age (GA)) or too late 108 (>32 weeks GA) booking at the first GDM visit, while in Figure 1 exclusion criteria was GA at 1st GDM visit not between 20-32 weeks (in text 24-32 weeks). Please explain.

2. In this study women were generally diagnosed with GDM at 24-28 weeks with OGTT. However, ADA guidelines suggest first screening before 15 weeks for early abnormal glucose metabolism, especially in those at risk.

3. Were any patients previously on metformin therapy or during pregnancy because of overweight or PCOS?

4. "Insulin treatment was initiated when glucose values were above targets more frequently than once per week for given period of the day, in accordance with Swiss guidelines". Please indicate that insulin treatment was initiated when fasting blood glucose was above...... and prandial above..... What sorts of insulins were used (in my country only detemir and aspart insulin in pregnancy).

5. In the Methods section you indicated that "The postpartum follow-up visits at both 6-8 weeks postpartum and 1-year postpartum included an assessment of the metabolic situation... The metabolic situation is not an optimal term in this condition because metabolic situation refers to serum lipids and blood pressure as well. Just indicate blood glucose and weight.

Results are clearly presented and discussed as well as discussion

Author Response

Dear reviewer, we have carefully looked at yourr comments and suggestions and have responded in our “response to reviewer’s comments” document, attached.

Reviewer 2 Report

Comments and Suggestions for Authors

The authors investigated weight trajectories, eating behaviors, and metabolic consequences in a cohort of women with gestational diabetes mellitus during pregnancy and postpartum according to pre-pregnancy BMI. The study results suggest an adaptive capacity to relatively rapid weight changes in the perinatal period that is less present with higher BMI.

The article is original and well written. Tables and figures are clear and the topic results relevant in the field of gestational diabetes mellitus.

I just have a few minor observations:

Figure 1: please correct evenly the spaces left before the numbers and after “n”.

Please report more clearly in the text which are the inclusion and exclusion criteria.

Please replace kg/m2 with kg/m2 throughout the text.

Line 242 the symbol used for normal pre-pregnancy BMI has a different color in the figure, please fix it.

Table 3 please eliminate the parenthesis in (kg).

Comments on the Quality of English Language

English language requires moderate editing.

Author Response

(The authors gave the same response as above.)

Reviewer 3 Report

Comments and Suggestions for Authors

The article focuses on the body weight trajectories, eating behaviors and some metabolic parameters during pregnancy and postpartum among women with GDM. The topic of the article is very important. However, there are a few major concerns that limit the validity of this work. Please address the following issues:

-        The title of the article suggests an assessment of metabolic parameters, but the study was limited to the glucose metabolic. I suggest changing the title (glycemic control instead of metabolic consequences) or to include other parameters in the results. It would be particularly interesting to consider triglyceride levels and other lipid parameters. 

-        The study covers the period from 2012 to 2022, including the period of the COVID lockdown. Some studies indicated that COVID-19 pandemic has affected the pregnancy outcome by increasing complications during pregnancy. Furthermore, public fear during the pandemic may have had a negative impact on pregnancy outcomes. Please explain the impact of the pandemic on the analyzed parameters.

-        Another doubt refers to the treatment of GDM with insulin. The authors mentioned (lines 128) that insulin treatment was indicated for some women. Insulin treatment can have a significant impact on weight gain during pregnancy. Please add information on insulin treatment, weight gain of pregnant women with GDM being treated vs not treated with insulin. 

-        Please precise what means very rarely (lines 152-153), as self-reported data do not have the same significance as measured data.

-        Please explain why was Swiss ethnicity separated from European ethnicity? Is it appropriate to combine European and North American ethnicities? In North America, there are 23 officially recognized independent states including Canada, United States, Mexico, Guatemala, ….

-        Eating habits were assessed using the IES-2 questionnaire, but this is only qualitative data. Quantitative data (energy and macronutrient intakes) are necessary to comprehensively assess the impact of nutrition on gestational body weight as well as postpartum body weight retention.

-        Also, physical activity during pregnancy throughout the period analyzed should be included as one of the factors influencing the results of the study.

-        The discussion provided within lines 348-380 is related to general mechanisms of overfeeding and adiposity in humans, but does not take into account the specificity of weight changes in pregnancies complicated by GDM.

-        Finally, the limitations of the study should be described in more detail.

In conclusion, the article requires major revisions before being published.

Author Response

(The authors gave the same response as above.)
